# Association between Body Mass Index and Hospital Outcomes for COVID-19 Patients: A Nationwide Study

**DOI:** 10.3390/jcm12041617

**Published:** 2023-02-17

**Authors:** Waleed Khokher, Saffa Iftikhar, Andrew Abrahamian, Azizullah Beran, Ziad Abuhelwa, Rakin Rashid, Hyder Ali, Sadik Khuder, Ragheb Assaly

**Affiliations:** 1Department of Internal Medicine, University of Toledo, Toledo, OH 43606, USA; 2Department of Medicine, University of Kansas St Francis Health, Topeka, KS 66606, USA; 3Department of Gastroenterology, Indiana University, Indianapolis, IN 47405, USA; 4Department of Internal Medicine, Mercy Catholic Medical Center, Darby, PA 19153, USA; 5Department of Internal Medicine, Rosalind Franklin University, McHenry Hospital, McHenry, IL 60050, USA; 6Department of Statistics and Public Health, University of Toledo, Toledo, OH 43606, USA; 7Department of Pulmonary and Critical Care Medicine, University of Toledo, Toledo, OH 43606, USA

**Keywords:** COVID-19, body mass index, mortality, intubation, hospital stay, nationwide study

## Abstract

Background: Coronavirus disease 2019 (COVID-19) caused significant morbidity and mortality worldwide. There is limited information describing the hospital outcomes of COVID-19 patients in regard to specific body mass index (BMI) categories. Methods: We utilized the Healthcare Cost and Utilization Project Nationwide Inpatient Sample (NIS) 2020 database to collect information on patients hospitalized for COVID-19 in the United States. Using the International Classification of Diseases, 10th revision, Clinical Modification (ICD-10-CM) coding system, adult patients (≥18 years of age) with a primary hospitalization for COVID-19 were identified. Adjusted analyses were performed to assess for mortality, morbidity, and resource utilization, and compare the outcomes among patients categorized according to BMI. Results: A total of 305,284 patients were included in this study. Of them, 248,490 had underlying obesity, defined as BMI ≥ 30. The oldest patients were observed to have BMI < 19, while youngest patients were in the BMI > 50 category. BMI < 19 category had the highest crude in-hospital mortality rate. However, after adjusted regression, patients with BMI > 50 (adjusted odds ratio (aOR) 1.63, 95% CI 1.48–1.79, *p*-value < 0.001) had the highest increased odds, at 63%, of in-hospital mortality compared to all other patients in the study. Patients with BMI > 50 also had the highest increased odds of needing invasive mechanical ventilation (IMV) and mortality associated with IMV compared to all other patient, by 37% and 61%, respectively. Obese patients were noted to have shorter average hospital length of stay (LOS), by 1.07 days, compared to non-obese patients, but there was no significant difference in average hospitalization charges. Conclusion: Among obese patients primarily hospitalized with COVID-19, those with BMI ≥ 40 had significantly increased rates of all-cause in-hospital mortality, need for IMV, mortality associated with IMV, and septic shock. Overall, obese patients had shorter average hospital LOS, however, did not have significantly higher hospitalization charges.

## 1. Introduction

Coronavirus disease 2019 (COVID-19), caused by severe acute respiratory syndrome coronavirus 2 (SARS-CoV-2) virus which was first discovered in December 2019, has gone on to become a worldwide pandemic causing significant morbidity and mortality [1,2]. Acute respiratory distress syndrome (ARDS) due to viral pneumonia is one of the leading causes of mortality among patients with COVID-19 [3]. ARDS occurs in 33% of hospitalized patients with COVID-19 and the average mortality rate among COVID-19 patients with ARDS is 39% (ranging from 13% to 73%) [3].

Several risk factors have been associated with increased mortality rates in people with a COVID-19 infection, specifically obesity [4,5]. Other risk factors include old age, male gender, African American race, the presence of chronic illness such diabetes, and chronic lung diseases such as lung cancer, chronic obstructive pulmonary disease (COPD), and asthma [4,5,6]. For patients with underlying obesity, it is a big concern when they develop COVID-19 as their risk of deterioration is greater and they have a poor prognosis if they develop ARDS and require invasive mechanical ventilation (IMV) [7].

There is no sufficient data to evaluate the mortality and morbidity associated with COVID-19 among specific body mass index (BMI) categories, specifically using Nationwide Inpatient Sample (NIS) databases. In this study, we evaluated the differences in baseline characteristics of the patients involved and all-cause in-hospital mortality in patients primarily admitted to the hospital for COVID-19. Our focus was to investigate if a specific BMI category conferred the greatest risk of mortality. Moreover, we investigated the morbidity and resource utilization outcomes during the hospital stay. To our knowledge, this is the largest study in the United States (US) utilizing NIS data on the outcomes of hospitalized COVID-19 patients focusing on a categorized BMI investigation and looking at risk of deterioration.

## 2. Materials and Methods

### 2.1. Data Source

This is a retrospective cohort study in which COVID-19 patients admitted to acute care hospitals in the US primarily due to a COVID-19 infection were identified using the Healthcare Cost and Utilization Project NIS 2020 database. Since the project utilized the NIS 2020 database, only patients from the year 2020 were included in the analysis. NIS is the largest all-payer and publicly available inpatient database in the United States which was developed and is maintained by the Agency for Healthcare Research and Quality. The NIS approach is described in detail in other sources [8]. Briefly summarized, NIS uses data from the American Hospital Association’s yearly hospital survey, hospitals are grouped according to ownership/control, bed number, teaching status, and geographic region. Then, data on patients’ demographics, diagnoses, and resource use are gathered from a random 20% sample of all patients within each stratum and collected and made available in the database. Each discharge is then weighted (weight = total number of discharges from all acute care hospitals in the US/number of discharges included in the 20% sample) to make the NIS nationally representative.

The NIS is the most comprehensive source of hospital data in the US, and it enables researchers to study healthcare delivery and patient outcomes [9]. This database is a discharge-level database that contains deidentified clinical and non-clinical data at the patient and hospital levels. As a result, multiple admissions for a single patient are considered separate and are entered separately into the database. Currently, the International Classification of Diseases, 10th revision, Clinical Modification (ICD-10-CM) coding system is used to code patient-level information.

### 2.2. Study Subjects

Adult patients (≥18 years old) with a primary diagnosis of a COVID-19 infection who were clearly categorized into to a certain BMI range using a secondary diagnosis were included in the study. Based on their BMI, the patients were divided into the following categories: BMI less than 19, BMI 20–29.9, BMI 30–39.9, BMI 40–49.9, and BMI greater than 50. These patients were also divided into obese (BMI ≥ 30) and non-obese (BMI < 30). Patients were excluded if they were less than 18 years of age, did not have a primary diagnosis of COVID-19, or did not have a secondary diagnosis which did not allow for them to be categorized via BMI. The ICD-10-CM diagnosis codes were used to extract data and are summarized in Appendix A. 

No review was necessary from the IRB at the University of Toledo as the study did not fall under the board’s classification as human subject research, and used de-identified patient information obtained from a publicly available national database.

### 2.3. Study Variables and Outcomes

The primary and secondary outcomes in this study separately looked at the five categories of BMI as described above and looked at obese patients (BMI ≥ 30) and compared them to the non-obese COVID-19 patients. Additionally, a separate analysis of the outcomes was conducted in which patients in the BMI 20–29.9 category were used as a reference to which all other BMI categories were compared. As people with BMI 20–29.9 are considered to have normal or overweight BMI, it was used as the reference BMI category.

The primary outcome was all-cause in-hospital mortality. Secondary outcomes were morbidity and resource utilization. Morbidity looked at the need for IMV, mortality associated with IMV, and occurrence of septic shock. Resource utilization looked at mean hospital length of stay (LOS) and mean hospitalization charges. Multiple potential confounders were collected and accounted for during the analysis, including patient’s age, gender, race, Charlson Comorbidity Index (CCI), and comorbidities including chronic heart failure, lung cancer, chronic pulmonary disease, diabetes mellitus, chronic renal failure, and smoking.

Patient status at discharge was coded directly in the NIS database, whether they were discharged alive. Septic shock, mechanical ventilation, and comorbidities (congestive heart failure, lung cancer, chronic pulmonary disease, diabetes mellitus, chronic renal failure, and smoking) were identified using the appropriate ICD-10-CM codes (Appendix A). The hospital LOS, hospitalization charges, and patients’ demographics were directly obtained from the NIS database.

### 2.4. Statistical Analysis

Data analysis was performed using STATA (IC-17.0 version, STATA Corp; College Station, TX, USA), which facilitated analyses to produce nationally representative unbiased results, variance estimates and *p*-values. Weighting of patient-level observations was implemented to obtain estimates for the entire population of hospitalized patients with COVID-19 in the US. The Rao-Scott chi-square test was used for categorical variables and Student’s t-test was used for continuous variables. Both univariate and multivariate regression models were created with the help of STATA-17.0 (College Station, TX, USA). If baseline characteristics were noted to be significantly different between BMI categories, multivariate regression models were used to adjust the results for potential confounders. Logistic regression was used for binary outcomes (all-cause in-hospital mortality, need for IMV, morality associated with IMV, and occurrence of septic shock) and linear regression was used for continuous outcomes (hospital LOS and total hospitalization charges). All *p*-values were two sided, and a value of 0.05 was the threshold for statistical significance.

## 3. Results

### 3.1. Patients Characteristics

Figure 1 shows the flow diagram for the selection of patients included in the study. A total of 305,284 patients were included in this study. Table 1 summarizes the baseline patient characteristics. All the baseline patient characteristics including age, gender, race, all comorbidities, and CCI were significantly different among all categories of BMI. Thus, all these characteristics would need to be adjusted for via multivariate regression. The overall mean age of all the subjects in the study was 60.1 (95% CI 59.9–60.3), with females representing 53.6% (95% CI 52.1–53.1) of the subjects. The oldest patients were those in the BMI < 19 subcategory, and the youngest patients were those in the BMI > 50 category. Additionally, patients in the BMI < 19 subcategory also had the highest proportion of patients with a CCI ≥ 3, followed by patients in BMI category 20–29.9, which means these patients had more chronic comorbidities and a higher 10-year mortality rate at baseline [10]. Table 2 summarizes the results of the overall crude outcomes in each BMI category.

### 3.2. Obese Versus Non-Obese Analysis

Table 3 summarizes regression analysis for the comparison of obese patients in which patients from BMI categorizes 30–39.9, 40–49.9 and >50 were combined and compared to non-obese patients in the BMI categories < 19 and 20–29.9. As there were significant differences in the baseline characteristics of the patients in each BMI category, multivariate regression was performed which adjusted for patients’ age, gender, race, comorbidities, and CCI. Multivariate regression was used to generate adjusted odds ratios (aOR) and adjusted mean differences (aMD).

Multivariate regression showed that in-hospital mortality was significantly higher in obese patients by 10% (aOR 1.10, 95% CI 1.02–1.19, *p*-value = 0.011). Need for IMV (aOR 1.37, 95% CI 1.26–1.50 *p*-value < 0.001), mortality associated with IMV (aOR 1.36, 95% CI 1.17–1.57, *p*-value < 0.001), and occurrence of septic shock (aOR 1.38, 95% CI 1.20–1.59, *p*-value < 0.001) were also significantly higher among obese patients, by 37%, 36%, and 38%, respectively. Mean hospital LOS was significantly shorter among obese patients by 1.07 days (aMD −1.07 days, 95% CI −1.32 to −0.83 days, *p*-value < 0.001). Mean hospital charges were not significantly different between obese and non-obese patients (aMD −4320 USD, 95% CI −9151 to 509 USD, *p*-value = 0.080). 

### 3.3. Separate BMI Category-Based Analysis

#### All-Cause in-Hospital Mortality

The overall all-cause in-hospital mortality rate for the COVID-19 patients in the study was 10% (Table 2). The highest mortality rate was observed in patients with BMI < 19 (18%) followed by the patients in the BMI 20–29.9 category (12.8%). The lowest rate of in-hospital mortality was observed for patients in the BMI 30–39.9 category (8.6%). 

Table 4 shows the results of the adjusted multivariate logistic regression analysis in which each BMI category was compared to all the other categories combined. This multivariate regression showed that patients with BMI 40–49.9 (aOR 1.24, 95% CI 1.16–1.32, *p*-value < 0.001) and BMI > 50 (aOR 1.63, 95% CI 1.48–1.79, *p*-value < 0.001) had significantly increased odds of in-hospital mortality, by 24% and 63%, respectively. Patients with BMI 20–29.9 (aOR 0.87, 95% CI 0.80–0.94, *p*-value = 0.001) and BMI 30–39.9 (aOR 0.78, 95% CI 0.74–0.83, *p*-value < 0.001) had significantly decreased odds of in-hospital mortality, by 13% and 22%, respectively. Patients with BMI < 19 (aOR 1.07, 95% CI 0.96–1.19, *p*-value = 0.248) did not have significantly different odds of in-hospital mortality when compared to all the other patients. 

Table 5 shows the results of the multivariate regression in which the BMI category 20–29.9 was used as a reference to which all other BMI categories were compared, while again adjusting for all baseline characteristics. BMI categories < 19 (aOR 1.15, 95% CI 1.02–1.31, *p*-value = 0.025), 40–49.9 (aOR 1.41, 95% CI 1.28–1.56, *p*-value < 0.001), and >50 (aOR 1.90, 95% CI 1.68–2.16, *p*-value < 0.001) all had significantly increased odds of in-hospital mortality when compared to the reference category. The BMI 30–39.9 (aOR 1.02, 95% CI 0.93–1.11, *p*-value = 0.728) category showed no significant difference compared to patients with BMI 20–29.9.

### 3.4. Morbidity

#### 3.4.1. Need for Invasive Mechanical Ventilation

The overall need for invasive mechanical ventilation (IMV) for the COVID-19 patients in the study was 11.3% (Table 2). The highest IMV rate was observed in patients with BMI > 50 (13.9%) followed by the patients in the BMI 40–49.9 category (12.6%). The lowest rate of IMV was observed for patients in the BMI < 19 category (6.4). 

Multivariate logistic regression analysis was performed comparing each BMI category to all the patients in the other categories combined (Table 4). It showed that patients with BMI 40–49.9 (aOR 1.24, 95% CI 1.17–1.32, *p*-value < 0.001) and BMI > 50 (aOR 1.37, 95% CI 1.26–1.49, *p*-value < 0.001) had significantly increased odds of requiring IMV, by 24% and 37%, respectively. Patients with BMI < 19 (aOR 0.51, 95% CI 0.44–0.59, *p*-value < 0.001), BMI 20–29.9 (aOR 0.89, 95% CI 0.81–0.97, *p*-value < 0.010), and BMI 30–39.9 (aOR 0.88, 95% CI 0.83–0.93, *p*-value < 0.001) had significantly decreased odds of requiring IMV, by 49%, 11%, and 12%, respectively. 

Multivariate logistic regression comparing the other BMI categories to BMI 20–29.9 was performed (Table 5). The following categories had significantly increased odds of requiring IMV when compared to the reference category, BMI 30–39.9 (aOR 1.10, 95% CI 1.001–1.21, *p*-value = 0.049), BMI 40–49.9 (aOR 1.43, 95% CI 1.29–1.59, *p*-value < 0.001), and BMI > 50 (aOR 1.68, 95% CI 1.48–1.90, *p*-value < 0.001). However, patients with BMI < 19 (aOR 0.57, 95% CI 0.48–0.68, *p*-value < 0.001) showed a significant decrease in need for IMV. 

#### 3.4.2. Mortality Associated with Invasive Mechanical Ventilation

The overall mortality associated with IMV for COVID-19 patients in the study was 53.4% (Table 2). The highest mortality after IMV was observed in patients with BMI 20–29.9 (55.2%) and the lowest was observed for patients with BMI 30–39.9 (52.4%). 

Multivariate logistic regression comparing each BMI category to all the other patients in the other categories (Table 4) showed that patients with BMI 40–49.9 (aOR 1.17, 95% CI 1.04–1.31, *p*-value = 0.008) and BMI > 50 (aOR 1.61, 95% CI 1.37–1.89, *p*-value < 0.001) had a significantly increased mortality rate associated with IMV, by 17% and 61%, respectively. Patients with BMI < 19 (aOR 0.62, 95% CI 0.45–0.84, *p*-value = 0.002), BMI 20–29.9 (aOR 0.80, 95% CI 0.68–0.94, *p*-value = 0.006), and BMI 30–39.9 (aOR 0.83, 95% CI 0.75–0.93, *p*-value = 0.001) had significantly decreased mortality associated with IMV, by 38%, 20%, and 17%, respectively. 

Multivariate regression comparing the other BMI categories to BMI 20–29.9 (Table 5) showed that patients with BMI 40–49.9 (aOR 1.45, 95% CI 1.21–1.74, *p*-value < 0.001) and BMI > 50 (aOR 2.02, 95% CI 1.61–2.53, *p*-value < 0.001) had significantly increased mortality associated with IMV, by 45% and 102%, respectively. Patients with BMI < 19 (aOR 0.74, 95% CI 0.74–1.03, *p*-value = 0.078) and BMI 30–39.9 (aOR 1.13, 95% CI 0.96–1.33, *p*-value = 0.149) show no significant difference in regard to mortality associated with IMV. 

### 3.5. Occurrence of Septic Shock

The overall rate of septic shock for COVID-19 patients in the study was 3.7% (Table 2). The highest rate of septic shock was observed in patients with BMI 40–49.9 (4.2%) and the lowest was observed for patients with BMI < 19 (2.8%). 

Multivariate logistic regression comparing each BMI category to all other patients (Table 4) showed that patients with BMI 40–49.9 (aOR 1.37, 95% CI 1.24–1.51, *p*-value < 0.001) and BMI > 50 (aOR 1.24, 95% CI 1.07–1.43, *p*-value = 0.003) had significantly increased rate of septic shock, 37% and 24%, respectively. Patients with BMI < 19 (aOR 0.60, 95% CI 0.47–0.76, *p*-value < 0.001), BMI 20–29.9 (aOR 0.84, 95% CI 0.73–0.97, *p*-value = 0.020), and BMI 30–39.9 (aOR 0.86, 95% CI 0.79–0.94, *p*-value = 0.001) had significantly decreased rate of septic shock, by 40%, 16%, and 14%, respectively. 

Multivariate regression comparing BMI categories to the reference category of BMI 20–29.9 (Table 5) showed that patients with BMI 40–49.9 (aOR 1.59, 95% CI 1.33–1.89, *p*-value < 0.001) and BMI > 50 (aOR 1.61, 95% CI 1.31–1.97, *p*-value < 0.001) had significantly increased rate of septic shock, by 59% and 61%, respectively. Patients with BMI < 19 (aOR 0.70, 95% CI 0.54–0.90, *p*-value = 0.006) had a significant 30% decrease in rate of septic shock, and patients with BMI 30–39.9 (aOR 1.13, 95% CI 0.97–1.32, *p*-value = 0.107) showed no significant difference. 

### 3.6. Resource Utilization

#### 3.6.1. Hospital Length of Stay

The overall mean hospital LOS for COVID-19 patients in the study was 8.26 days (Table 2). The longest mean LOS was observed in patients with BMI < 19 and BMI 20–29.9, 9.79 and 9.75 days, respectively. The shortest mean LOS was seen in patients with BMI > 50, 8.04 days. 

Multivariate linear regression comparing each BMI category to all other patients (Table 4) showed that patients with BMI 30–39.9 (aMD −0.82 days, 95% CI −0.96 to −0.69 days, *p*-value < 0.001) had a significantly shorter hospital LOS by 0.82 days. Patients with BMI < 19 (aMD 0.66 days, 95% CI 0.22 to 1.10 days, *p*-value = 0.003), BMI 20–29.9 (aMD 1.04 days, 95% CI 0.77 to 1.31 days, *p*-value < 0.001), BMI 40–49.9 (aMD 0.17 days, 95% CI 0.01 to 0.34 days, *p*-value = 0.042), and BMI > 50 (aMD 0.42 days, 95% CI 0.19 to 0.65 days, *p*-value < 0.001) had significantly longer hospital LOS, by 0.66, 1.04, 0.17, and 0.42 days, respectively. 

Multivariate linear regression comparing BMI categories to the reference category of BMI 20–29.9 (Table 5) showed that patients with BMI 30–39.9 (aMD −1.33 days, 95% CI −1.60 to −1.07 days, *p*-value < 0.001), BMI 40–49.9 (aMD −0.79 days, 95% CI −1.10 to −0.48 days, *p*-value < 0.001), and BMI > 50 (aMD −0.58 days, 95% CI −0.93 to −0.22 days, *p*-value = 0.001) had significantly shorter hospital LOS, by 1.33, 0.79, and 0.58 days, respectively. Patients with BMI < 19 (aMD −0.25 days, 95% CI −0.73 to 0.22 days, *p*-value = 0.296) showed no significant difference in hospital LOS compared to patients with BMI 20–29.9. 

#### 3.6.2. Total Hospitalization Charges

The overall mean charges for hospitalization for COVID-19 patients in the study was 89,856 US dollars (USD) (Table 2). The highest mean charges were observed in patients with BMI 20–29.9 with 100,485 USD. The lowest mean charges were seen in patients with BMI 30–39.9 at 84,593 USD. 

Multivariate linear regression comparing each BMI category to all other patients (Table 3) showed that patients with BMI 30–39.9 (aMD −10,921 USD, 95% CI −13,647 to −8196 USD, *p*-value < 0.001) had significantly lower hospital charges by 10,921 USD. Patients with BMI 20–29.9 (aMD 6809 USD, 95% CI 1950 to 11,668 USD, *p*-value = 0.006), BMI 40–49.9 (aMD 5984 USD, 95% CI 2569 to 9398 USD, *p*-value < 0.001), and BMI > 50 (aMD 11,901 USD, 95% CI 6871 to 16,931 USD, *p*-value < 0.001) had significantly higher mean hospital charges, by 6809, 5984 and 11,901 USD, respectively, while patients with BMI < 19 (aMD −2977 USD, 95% CI −10,969 to 5015 USD, *p*-value = 0.465) showed no significant difference. 

Multivariate linear regression comparing each BMI category to the reference category of BMI 20–29.9 (Table 4) showed that patients with BMI < 19 (aMD −8477 USD, 95% CI −16,817 to −136 USD, *p*-value = 0.046) and BMI 30–39.9 (aMD −10,459 USD, 95% CI −15,440 to −5478 USD, *p*-value < 0.001) had significantly lower mean hospital charges, by 8477 and 10,459 USD, respectively. Patients with BMI 40–49.9 (aMD 296 USD, 95% CI −5684 to 6276 USD, *p*-value = 0.923) and BMI > 50 (aMD 6793 USD, 95% CI −289 to 13,875 USD, *p*-value = 0.060) showed no significant difference compared to patients with BMI 20–29.9. 

## 4. Discussion

Utilizing the NIS database found that COVID-19 patients with obesity had significantly higher rates of all-cause in-hospital mortality compared to non-obese patients. After adjusted analysis, the presence of obesity in hospitalized COVID-19 patients was associated with a significant increase by 10% in the odds of in-hospital mortality. These results were in line with the study by Palaiodimos et al. [11], which found that obesity leads to a four-fold increase in in-hospital mortality. Similarly, when comparing obese patients to a normal BMI range (20–29.9), a study conducted in the United Arab Emirates including 1796 by Hafez et al. [12] showed a non-significant increase in mortality for patients with Class III obesity (BMI ≥ 40). Another study conducted in Japan by Lee et al. [13], which included 1837 patients, did not show significant differences in morality among the BMI classes when compared to each other. However, our study showed a significant increase in mortality for patients with BMI ≥ 40, both compared to other BMI categories and specifically compared to normal BMI of 20–29.9, likely owing to the increased power of our investigation. Our results were similar to a study conducted in Italy by Graziano et al. [14], who found that an increased waist circumference and abdominal adiposity were associated with an increased need for IMV. We also observed an increase of 37% for odds of requiring IMV, 36% for odds of mortality associated with IMV, and 38% for odds of developing septic shock among obese patients. These results were validated by a study by Simonnet et al. [15], which found a seven-fold increase in IMV in obese COVID-19 patients. Obese patients had on average a shorter hospital LOS by 1.07 days, but no significant difference in mean hospital charges compared to non-obese patients. A major part of our study was looking at the specific BMI categories and comparing mortality, morbidity, and resource utilization. To our knowledge, this is the largest study in the US investigating the role of specific BMIs in COVID-19 patients utilizing the NIS database. 

After adjusting for the baseline characteristics summarized in Table 1, we found that patients with BMI > 50 had significantly higher odds of all-cause in-hospital mortality both when compared to all other study patients and when specifically compared to patients with BMI 20–29.9, a significant increase by 63% and 90%, respectively. As expected, patients with BMI 20–29.9 had significantly lower rates of in-hospital mortality. Similarly, patients with BMI 30–39.9 also had lower odds of in-hospital mortality when compared to all the other BMI categories, but most importantly mortality was not significantly different when compared to BMI 20–29.9. This may be because, as the data suggests, there is a cutoff and patients who have BMI ≥ 40 have a significantly elevated in-hospital mortality rate. This is supported by the findings of a study by Hendren et al. [16] which looked at a registry of patients with chronic heart disease and COVID-19 and found that those with BMI ≥ 40 had significantly higher in-hospital mortality. This is clearly demonstrated in our data as patients who crossed into the BMI 40–49.9 and BMI > 50 categories, they had significantly elevated in-hospital mortality rates both when compared to all other BMI categories combined and when compared to BMI 20–29.9, a significant increase by 24% and 41%, respectively. This trend was also observed when looking at the need for IMV, mortality associated with IMV, and septic shock. Evidence for this cutoff point for worse hospital outcomes is also supported by our findings: that patients with BMI 30–39.9 had lower odds of IMV when compared to all other BMI categories and only mildly elevated odds when compared to the BMI 20–29.9 category patients. Similar trends were observed for mortality associated with IMV and septic shock, as patients in the BMI 20–29.9 and 30–39.9 category showed no significant differences and had similar odds when compared to all other patients. 

Conversely, lower BMIs were not necessarily associated with better hospital outcomes. Patients with BMI < 19 had significantly higher in-hospital mortality compared to all other patients. As seen in Table 1, patients with BMI < 19 had the oldest patient population with the highest portion of patients with CCI ≥ 3. This likely represents a disproportionate number of baseline unhealthy patients that are more prone to death with severe COIVD-19 infection. Additionally, it is important to note that adult patients with BMI < 19 are also likely to be malnourished [17]. Thus, these patients are more likely to succumb to worse in-hospital outcomes due to decreased nutritional status [18]. However, patients with BMI < 19 did have lower odds of requiring IMV and septic shock both when compared to all other BMI categories combined and BMI 20–29.9 specifically. A plausible explanation for this could be because the patients in the BMI < 19 category were not only significantly older; they were also not very healthy at baseline, as evidenced by CCI ≥ 3. Thus, perhaps the patients, family members, and the medical team chose to pursue comfort methods rather than aggressive medical treatment. Evidence for this is suggested in Table 2, as a disproportionately lower number of patients with BMI < 19 received IMV at 6.4% and a diagnosis of septic shock at 2.8%, even though they had the highest rate of crude mortality at 18%.

Resource utilization is an important part of hospital management, and so we investigated the average hospital LOS and charges associated with COVID-19 admissions. Multivariate regression showed that the mean hospital LOS was significantly longer in the categories of BMI < 19, BMI 20–29.9, BMI 40–49.9, and BMI > 50 and significantly shorter among patients with BMI 30–39.9 when compared to all the other patients. When compared to BMI 20–29.9 as a reference, all categories except BMI < 19 showed a statistically shorter average hospital LOS, while patients with BMI < 19 did not show a significant difference. A reason for these findings can be that in patients with COVID-19 pneumonia, escalation of care to IMV is delayed and patients are often observed with awake-prone positioning and non-invasive ventilation (NIV) [19,20]. This delay in IMV exists as the mortality associated with IMV in COVID-19 patients is notoriously high [21]. As patients become heavier, it becomes more challenging to prone them while awake due to discomfort and body habitus. As a result, these patients may need IMV sooner as an improvement in ventilation–perfusion (V/Q) mismatch cannot be achieved via proning [22]. Thus, higher mortality earlier during hospitalization leads to shorter hospital LOS for obese patients with COVID-19 ARDS. Conversely, lighter patients with BMI 20–29.9 can be observed for longer periods of time while awake-proning and on NIV while in the hospital setting, leading to longer average hospital LOS [19]. 

Hospital LOS can often help to define the hospitalization charges. We found that the mean hospital charges were significantly higher in the BMI 20–29.9, 40–49.9, and BMI > 50 categories and significantly lower among patients in the BMI 30–39.9 categories when each category was compared to all the other patients. When compared to patients with BMI 20–29.9, those with BMI 30–39.9 had significantly lower hospital charges, while those with BMI 40–49.9 and BMI > 50 showed no significant difference. This is likely related to a combination of increased hospital LOS and earlier need for IMV. Patients with BMI 40–49.9 and >50 likely require IMV earlier. IMV warrants intensive care unit (ICU) admission which significantly increases costs of hospitalization [23]. As for patients with BMI 20–29.9, they could be observed without IMV while being awake-proned in the hospital setting, but this would increase their hospital LOS and therefore hospitalization cost. 

There are several limitations of our study that should be acknowledged. First, the retrospective nature of the study does not allow for conclusive evidence as patients could not be randomized and regression models were utilized to account for confounders. Thus, the risk of residual confounding remains. Second, the NIS database is a claims-based database, and as such, it is vulnerable to incorrect datapoints or diagnoses based on clinician judgement. Third, the lack of certain important patient-level data did not allow for controlling for possible variations among patients because other medications used for COVID-19 treatment might have been different and not standardized between patients treated at different facilities. Fourth, a lack of clinical patient-level data did not allow for us to investigate clinical decision making. Thus, we had to formulate certain assumptions based on best medical practices. Fifth, the data for this study were obtained from a US-based national database, and so the results may not be generalizable to other countries because of different treatment standards. Lastly, this was the first time COVID-19 national data were compiled in the NIS database and treatment standards have changed since the pandemic first started. Thus, the results may vary significantly for current COVID-19 hospitalization as treatment standards have changed. Despite the limitations, our study has significant strengths. We conducted our study using a nationwide database which gave us the ability to include over 305,000 patients. To our knowledge, this is the first US-based nationwide study conducted looking at the role of specific BMI in COVID-19 outcomes.

## 5. Conclusions

Our study showed that the obese patients hospitalized with COVID-19 had a higher rate of all-cause in-hospital mortality, need for IMV, mortality associated with IMV, and septic shock. Obese patients also had shorter average hospital LOS by roughly 1 day compared to non-obese patients, however, there was no significant difference in hospitalization charges. Among obese patients, those with BMI ≥ 40 were the ones suffering from increased in-hospital mortality, increase rated of IMV, IMV associated mortality, and septic shock, whereas patients with BMI 30–39.9 had similar outcomes to those with BMI 20–29.9, and both groups had lower rates of in-hospital mortality compared to patients with BMI < 19.

## Figures and Tables

**Figure 1 jcm-12-01617-f001:**
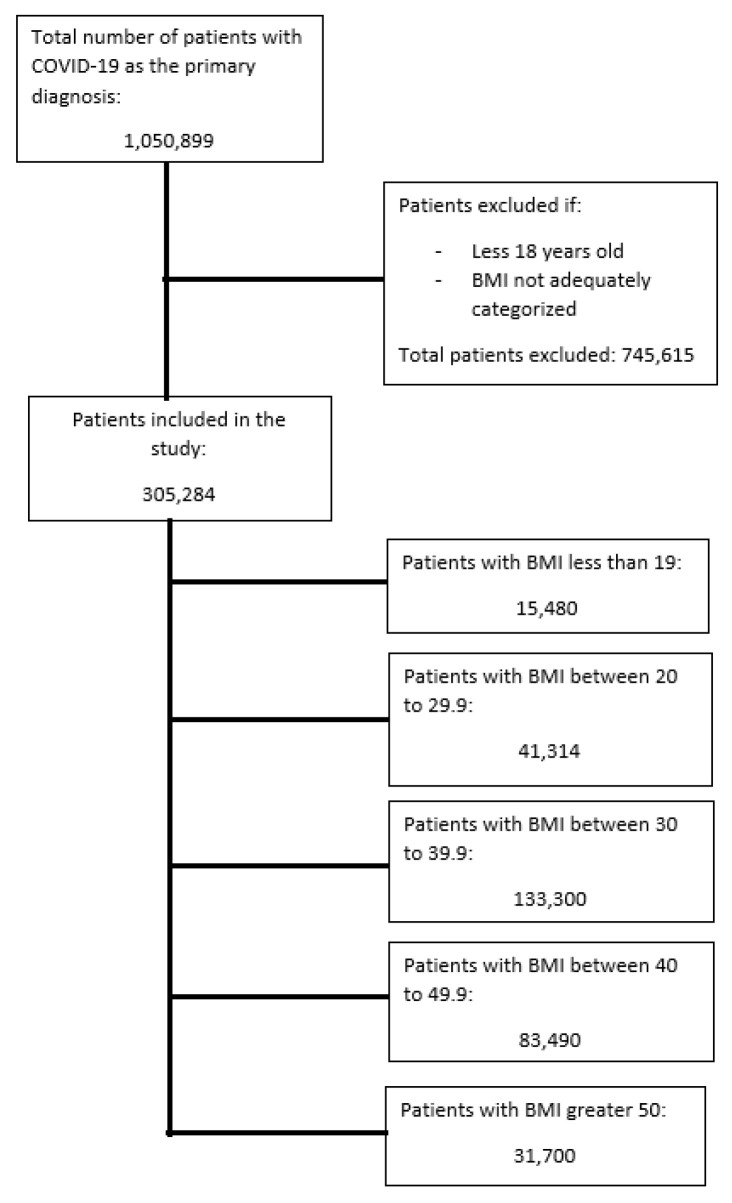
Selection of patients in the study.

**Table 1 jcm-12-01617-t001:** Patient baseline characteristics.

		Non-Obese (n = 56,794)		Obese (n = 248,490)			
		BMI < 19 (n = 15,480)	BMI 20–29.9 (n = 41,314)	BMI 30–39.9 (n = 133,300)	BMI 40–49.9 (n = 83,490)	BMI > 50 (n = 31,700)	*p*-Value
Age (year) mean (95% CI)		77.2 (76.7–77.7)	69.8 (69.4–70.2)	60.0 (59.8–60.2)	55.9 (55.7–56.2)	50.4 (50.1–50.8)	<0.001 *
Females, n (%)		8394 (54.5)	17,859 (43.4)	63,184 (47.5)	49,845 (59.7)	21,000 (66.2)	<0.001 *
Race n (%)	White	9044 (58.4)	20,479 (49.6)	64,169 (48.1)	40,924 (49.0)	14,470 (45.6)	<0.001 *
	Black	2895 (18.7)	6755 (16.4)	25,335 (19.0)	20,960 (25.1)	9429 (29.7)	
	Hispanic	1629 (10.5)	9144 (22.1)	31,044 (23.3)	14,900 (17.8)	5205 (16.4)	
	Asian or Pacific Islander	859 (5.5)	1519 (3.7)	2490 (1.9)	989 (1.2)	484 (1.5)	
	Native American	105 (0.7)	264 (0.6)	1459 (1.1)	1009 (1.2)	389 (1.2)	
	Other	479 (3.1)	1789 (4.3)	4754 (3.6)	2300 (2.8)	810 (2.6)	
	Missing	374 (2.4)	1229 (3.0)	3760 (2.8)	2345 (2.8)	910 (2.9)	
Comorbidities	Chronic pulmonary disease, n (%)	5089 (32.9)	9199 (22.3)	31,719 (23.8)	24,945 (29.9)	10,925 (34.5)	<0.001 *
	Smoking, n (%)	4795 (31.0)	12,134 (29.4)	37,904 (28.4)	20,709 (24.8)	7285 (23.0)	<0.001 *
	Chronic kidney disease, n (%)	3884 (25.1)	9239 (22.4)	22,654 (17.0)	14,920 (17.9)	4954 (15.6)	<0.001 *
	Diabetes mellitus, n (%)	3579 ()	16,419 ()	62,170 ()	41,910 ()	15,640 ()	<0.001 *
	Chronic heart failure, n (%)	3274 (21.1)	6444 (15.6)	17,534 (13.2)	15,259 (18.3)	6790 (21.4)	<0.001 *
	Lung cancer, n (%)	235 (1.5)	315 (0.8)	309 (0.2)	114 (0.1)	14 (0.04)	<0.001 *
Charlson comorbidity index n (%)	0	1654 (10.7)	9174 (22.2)	38,449 (28.8)	21,130 (25.3)	7930 (25.0)	<0.001 *
	1	4104 (26.5)	11,405 (27.6)	40,730 (30.6)	25,595 (30.7)	9715 (30.6)	
	2	3179 (20.5)	7229 (17.5)	21,639 (16.2)	14,270 (17.1)	5805 (18.3)	
	≥3	6449 (41.7)	13,374 (32.4)	32,194 (24.2)	22,435 (26.9)	8249 (26.0)	

Abbreviations: CI = confidence interval, n = number of patients; * Statistical significance (*p*-value < 0.05).

**Table 2 jcm-12-01617-t002:** Crude unadjusted outcomes of the study.

		All COVID-19 Patients in the Study, n (%)	BMI < 19, n (%)	BMI 20–29.9, n (%)	BMI 30–39.9, n (%)	BMI 40–49.9, n (%)	BMI > 50, n (%)
Mortality	All-cause in-hospital mortality	30,612 (10.0)	2764 (18.0)	5284 (12.8)	11,465 (8.6)	7964 (9.5)	3135 (9.9)
Morbidity	Need for IMV	34,404 (11.3)	994 (6.4)	4419 (10.7)	14,024 (10.5)	10,554 (12.6)	4410 (13.9)
	Mortality associated with IMV	18,364 (53.4)	524 (52.7)	2439 (55.2)	7349 (52.4)	5644 (53.5)	2405 (54.5)
	Septic shock	11,419 (3.7)	429 (2.8)	1565 (3.8)	4664 (3.5)	3519 (4.2)	1240 (3.9)
Resource Utilization	Mean hospital LOS (days)	8.26	9.79	9.75	7.79	8.10	8.04
	Mean hospitalization charges (USD)	89,856	88,627	100,485	84,593	91,420	95,823

Abbreviations: BMI = body mass index, IMV = invasive mechanical ventilation, LOS = length of stay, USD = United States Dollar.

**Table 3 jcm-12-01617-t003:** Regression outcomes of obese patients.

		Univariate	*p*-Value ^#^	Multivariate	*p*-Value ^#^
Mortality	All-cause in-hospital mortality, OR (95% CI)	0.60 (0.56–0.64)	<0.001 *	1.10 (1.02–1.19)	0.011 *
Morbidity	Need for IMV, OR (95% CI)	1.25 (1.16–1.35)	<0.001 *	1.37 (1.26–1.50)	<0.001 *
	Mortality Associated with IMV, OR (95% CI)	0.94 (0.82–1.07)	0.338	1.36 (1.17–1.57)	<0.001 *
	Septic Shock, OR (95% CI)	1.08 (0.96–1.22)	0.214	1.38 (1.20–1.59)	<0.001 *
Resource Utilization	Mean Hospital LOS, days (95% CI)	−1.80 (−2.04 to −1.57)	<0.001 *	−1.07 (−1.32 to −0.83)	<0.001 *
	Mean Hospitalization Charges, USD (95% CI)	−8698 (−13,157 to −4239)	<0.001 *	−4320 (−9151 to 509)	0.080

Abbreviations: BMI = body mass index, CI = confidence interval, IMV = invasive mechanical ventilation, OR = odds ratio, LOS = length of stay, USD = United States Dollar; ^#^
*p*-value to determine significance of difference between obese and non-obese patients; * Statistical significance (*p*-value < 0.05).

**Table 4 jcm-12-01617-t004:** Regression outcomes of separate body mass index categories.

		BMI Category	Univariate	*p*-Value ^#^	Multivariate	*p*-Value ^#^
Mortality	All-cause in-hospital mortality, OR (95% CI)	BMI < 19	2.06 (1.87–2.28)	<0.001 *	1.07 (0.96–1.19)	0.248
		BMI 20–29.9	1.39 (1.29–1.50)	<0.001 *	0.87 (0.80–0.94)	0.001 *
		BMI 30–39.9	0.75 (0.71–0.80)	<0.001 *	0.78 (0.74–0.83)	<0.001 *
		BMI 40–49.9	0.93 (0.87–0.99)	0.019 *	1.24 (1.16–1.32)	<0.001 *
		BMI > 50	0.98 (0.90–1.07)	0.660	1.63 (1.48–1.79)	<0.001 *
Morbidity	Need for IMV, OR (95% CI)	BMI < 19	0.53 (0.46–0.62)	<0.001 *	0.51 (0.44–0.59)	<0.001 *
		BMI 20–29.9	0.94 (0.86–1.02)	0.159	0.89 (0.81–0.97)	0.010 *
		BMI 30–39.9	0.88 (0.83–0.93)	<0.001 *	0.88 (0.83–0.93)	<0.001 *
		BMI 40–49.9	1.20 (1.13–1.27)	<0.001 *	1.24 (1.17–1.32)	<0.001 *
		BMI > 50	1.31 (1.21–1.41)	<0.001 *	1.37 (1.26–1.49)	<0.001 *
	Mortality associated with IMV, OR (95% CI)	BMI < 19	0.98 (0.74–1.30)	0.905	0.62 (0.45–0.84)	0.002 *
		BMI 20–29.9	1.09 (0.94–1.26)	0.272	0.80 (0.68–0.94)	0.006 *
		BMI 30–39.9	0.94 (0.85–1.03)	0.192	0.83 (0.75–0.93)	0.001 *
		BMI 40–49.9	1.01 (0.91–1.12)	0.843	1.17 (1.04–1.31)	0.008 *
		BMI > 50	1.05 (0.92–1.21)	0.459	1.61 (1.37–1.89)	<0.001 *
	Septic shock, OR (95% CI)	BMI < 19	0.73 (0.59–0.91)	0.006 *	0.60 (0.47–0.76)	<0.001 *
		BMI 20–29.9	1.02 (0.89–1.17)	0.763	0.84 (0.73–0.97)	0.020 *
		BMI 30–39.9	0.89 (0.81–0.97)	0.006 *	0.86 (0.79–0.94)	0.001 *
		BMI 40–49.9	1.19 (1.09–1.31)	<0.001 *	1.37 (1.24–1.51)	<0.001 *
		BMI > 50	1.05 (0.92–1.20)	0.465	1.24 (1.07–1.43)	0.003 *
Resource Utilization	Mean hospital LOS, days (95% CI)	BMI < 19	1.61 (1.21 to 2.02)	<0.001 *	0.66 (0.22 to 1.10)	0.003 *
		BMI 20–29.9	1.72 (1.46 to 1.99)	<0.001 *	1.04 (0.77 to 1.31)	<0.001 *
		BMI 30–39.9	−0.82 (−0.96 to −0.69)	<0.001 *	−0.82 (−0.96 to −0.69)	<0.001 *
		BMI 40–49.9	−0.21 (−0.38 to −0.47)	0.012 *	0.17 (0.01 to 0.34)	0.042 *
		BMI > 50	−0.24 (−0.47 to −0.02)	0.034 *	0.42 (0.19 to 0.65)	<0.001 *
	Mean hospitalization charges, USD (95% CI)	BMI < 19	−1295 (−8385 to 5795)	0.720	−2977 (−10,969 to 5015)	0.465
		BMI 20–29.9	12,296 (7279 to 17,312)	<0.001 *	6809 (1950 to 11,668)	0.006 *
		BMI 30–39.9	−9354 (−12,186 to −6522)	<0.001 *	−10,921 (−13,647 to −8196)	<0.001 *
		BMI 40–49.9	2153 (−1166 to 5472)	0.204	5984 (2569 to 9398)	0.001 *
		BMI > 50	6660 (1585 to 11,736)	0.010 *	11,901 (6871 to 16,931)	<0.001 *

Abbreviations: BMI = body mass index, CI = confidence interval, IMV = invasive mechanical ventilation, OR = odds ratio, LOS = length of stay, USD = United States Dollar. ^#^
*p*-value to determine significance of difference between each BMI category and rest of the BMI categories combined. * Statistical significance (*p*-value < 0.05).

**Table 5 jcm-12-01617-t005:** Regression outcomes of comparing the body mass index categories to body mass index 20–29.9 as a reference.

		BMI Category	Multivariate	*p*-Value ^#^
Mortality	All-cause in-hospital mortality, OR (95% CI)	BMI 20–29.9	Reference	Reference
		BMI < 19	1.15 (1.02–1.31)	0.025 *
		BMI 30–39.9	1.02 (0.93–1.11)	0.728
		BMI 40–49.9	1.41 (1.28–1.56)	<0.001 *
		BMI > 50	1.90 (1.68–2.16)	<0.001 *
Morbidity	Need for IMV, OR (95% CI)	BMI 20–29.9	Reference	Reference
		BMI < 19	0.57 (0.48–0.68)	<0.001 *
		BMI 30–39.9	1.10 (1.001–1.21)	0.049 *
		BMI 40–49.9	1.43 (1.29–1.59)	<0.001 *
		BMI > 50	1.68 (1.48–1.90)	<0.001 *
	Mortality associated with IMV, OR (95% CI)	BMI 20–29.9	Reference	Reference
		BMI < 19	0.74 (0.53–1.03)	0.078
		BMI 30–39.9	1.13 (0.96–1.33)	0.149
		BMI 40–49.9	1.45 (1.21–1.74)	<0.001 *
		BMI > 50	2.02 (1.61–2.53)	<0.001 *
	Septic shock, OR (95% CI)	BMI 20–29.9	Reference	Reference
		BMI < 19	0.70 (0.54–0.90)	0.006 *
		BMI 30–39.9	1.13 (0.97–1.32)	0.107
		BMI 40–49.9	1.59 (1.33–1.89)	<0.001 *
		BMI > 50	1.61 (1.31–1.97)	<0.001 *
Resource Utilization	Mean hospital LOS, days (95% CI)	BMI 20–29.9	Reference	Reference
		BMI < 19	−0.25 (−0.73 to 0.22)	0.296
		BMI 30–39.9	−1.33 (−1.60 to −1.07)	<0.001 *
		BMI 40–49.9	−0.79 (−1.10 to −0.48)	<0.001 *
		BMI > 50	−0.58 (−0.93 to −0.22)	0.001 *
	Mean hospitalization charges, USD (95% CI)	BMI 20–29.9	Reference	Reference
		BMI < 19	−8477 (−16,817 to −136)	0.046 *
		BMI 30–39.9	−10,459 (−15,440 to −5478)	<0.001 *
		BMI 40–49.9	296 (−5684 to 6276)	0.923
		BMI > 50	6793 (−289 to 13,875)	0.060

Abbreviations: BMI = body mass index, CI = confidence interval, IMV = invasive mechanical ventilation, OR = odds ratio, LOS = length of stay, USD = United States Dollar. ^#^
*p*-value to determine significance of difference between each BMI category and the reference category of BMI 20–29.9. * Statistical significance (*p*-value < 0.05).

## Data Availability

All datasets created for the purposes of the manuscript can be obtained from the corresponding author upon request.

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
