# Peer review of "Association between Body Mass Index and Hospital Outcomes for COVID-19 Patients: A Nationwide Study"

_jcm, 2023, doi:10.3390/jcm12041617_

Round 1

Reviewer 1 Report

In this paper based of administrative data the authors describe how body mass index associates with worse  hospital outcomes in obese COVID-19 patients.

The study is of interest and well structured.

Only some minor points should be addressed.

I would add some general epidemiogical data about the whole population (age and sex) in the patients characteristics.

I would specify the recruting period in the data source. The only mention to time is in present in the Healthcare Cost and Utilization Project NIS 2020 database.

In table 1 a graphical modification should be addeded to improve reading, separating Non obese from obese patients, such as vertical line.

Even though the paper deals with the US, I would add  in the discussion a short comparison of the main findings (association with BMI and other determinants of mortality) with  studies carried out in other countries, notably Europe which was heavily and initially affected by the pandemic.

I add some references.

Hafez W, Abdelshakor M, Kishk S, Gebril A, Gador M, Osman S, Abuelsaoud HM, Abdelrahman A. Body Mass Index and Clinical Outcomes in Adult COVID-19 Patients of Diverse Ethnicities. Healthcare (Basel). 2022 Dec 19;10(12):2575

Lenti MV, Uderzo S, Rossi CM, Melazzini F, Klersy C, Ferretti VV, Di Sabatino A. Determinants of COVID-19-related mortality in an internal medicine setting. Intern Emerg Med. 2022 Oct;17(7):2169-2173.

Graziano E, Peghin M, De Martino M, De Carlo C, Da Porto A, Bulfone L, et al; GIRA-COVID study group. The impact of body composition on mortality of COVID-19 hospitalized patients: A prospective study on abdominal fat, obesity paradox and sarcopenia. Clin Nutr ESPEN. 2022 Oct;51:437-444.

Lee H, Chubachi S, Namkoong H, Tanaka H, Otake S, Nakagawara K, et. al; Japan COVID-19 Task Force. Effects of mild obesity on outcomes in Japanese patients with COVID-19: a nationwide consortium to investigate COVID-19 host genetics. Nutr Diabetes. 2022 Aug 9;12(1):38.

Scheffler M, Genton L, Graf CE, Remuinan J, Gold G, Zekry D, et al  Prognostic Role of Subcutaneous and Visceral Adiposity in Hospitalized Octogenarians with COVID-19. J Clin Med. 2021 Nov 24;10(23):5500.

  1.  

Reviewer 2 Report

This is a well-designed and methodologically sound retrospective study, on a large database of hospitalized COVID patients.

Some observations:

-       Please explain the ethical approval or waiving of such (if it is the case) for this study; there is no reference in the text to an Institutional Review Board for approval of this study.

-       The references do not respect the style recommended by the journal

-       In the discussions – rows 386-401 – this paragraph draws conclusions that are not reflected in the data; there are several suppositions which are not supported by contemporary management of ARDS patients, such as that BMI 30-39.9 patients are less frequently proned than lighter patients, which leads to higher early mortality; also, there are several inferences on awake proning which are not supported by collected data; authors should refrain from drawing conclusions not deriving from data

-       The cited reference (Langer) does not address this supposition and should not be used to support it; the whole paragraph should be rephrased.

-       Also, in the previous paragraph, one of the references (Palmon) is an old study in anesthesia (not ICU ARDS patients) that does not support the thesis (difficulty of proning obese ICU patients); please use adequate reference or remove argument.

-       In general, one of the limitations which the authors recognise should be taken into considerations when inferring on the results – “a lack of clinical patient-level data did not allow for us to investigate clinical decision making”

Round 2

Reviewer 2 Report

The authors have addressed all the suggestions, I have nothing more to add